# Oxidative Stress Is Associated with Overgrowth in Drosophila *l(3)mbt* Mutant Imaginal Discs

**DOI:** 10.3390/cells11162542

**Published:** 2022-08-16

**Authors:** Paula Climent-Cantó, Cristina Molnar, Paula Santabárbara-Ruiz, Cristina Prieto, Josep F. Abril, Florenci Serras, Cayetano Gonzalez

**Affiliations:** 1Departament de Genètica, Microbiologia i Estadística, Facultat de Biologia, Universitat de Barcelona, 08028 Barcelona, Spain; 2Institute for Research in Biomedicine (IRB Barcelona), The Barcelona Institute of Science and Technology, Baldiri Reixac, 10, 08028 Barcelona, Spain; 3Institut de Biomedicina de la Universitat de Barcelona (IBUB), 08028 Barcelona, Spain; 4Catalan Institution for Research and Advanced Studies (ICREA), 08010 Barcelona, Spain

**Keywords:** drosophila cancer model, ROS, oxidative stress, hyperplasia

## Abstract

The loss-of-function conditions for an *l(3)malignant brain tumour* (*l(3)mbt*) in larvae reared at 29 °C results in malignant brain tumours and hyperplastic imaginal discs. Unlike the former that have been extensively characterised, little is known about the latter. Here we report the results of a study of the hyperplastic *l(3)mbt* mutant wing imaginal discs. We identify the *l(3)mbt* wing disc tumour transcriptome and find it to include genes involved in reactive oxygen species (ROS) metabolism. Furthermore, we show the presence of oxidative stress in *l(3)mbt* hyperplastic discs, even in apoptosis-blocked conditions, but not in *l(3)mbt* brain tumours. We also find that chemically blocking oxidative stress in *l(3)mbt* wing discs reduces the incidence of wing disc overgrowths. Our results reveal the involvement of oxidative stress in *l(3)mbt* wing discs hyperplastic growth.

## 1. Introduction

Research in *Drosophila* has led to important discoveries on the molecular mechanisms that govern tumour induction and progression [1,2]. One of the experimental tumour models reported in *Drosophila* is based on loss-of-function conditions for a *lethal (3) malignant brain tumour* (*l(3)mbt*). The tumour suppressor *l(3)mbt* encodes a conserved transcriptional regulator that is ubiquitously expressed in *Drosophila*. L(3)mbt harbours three MBT repeats and a zinc finger domain [3], binds to insulator sequences [4], and has a role in histone compaction [5,6]. Extensive biochemical studies show that L(3)mbt interacts with the dREAM complex, as well as the L(3)mbt-interacting (LINT) complex, which control gene expression and repress developmental genes [7,8]. L(3)mbt has been shown to repress germline genes in somatic tissues, as well as testis-specific and neuronal genes in the female germline, and also to be essential for repressing SWH target genes [4,9,10,11].

Mutations affecting the Drosophila *l(3)mbt* gene cause malignant brain tumours (hence referred to as mbt tumours) that originate in the neuroepithelial regions of the larval brain lobes [4,12]. Upon allografting, mbt tumours grow, invade the abdomen, and kill the host. Indeed, mbt tumours can undergo infinite rounds of allografting and are, therefore, immortal [10,12].

Gene expression profiling of brain mbt tumours reveals a tumour signature that is enriched in genes that are normally expressed only in the germline, some of which are essential for mbt tumour growth [10,13]. Notably, individuals carrying the *l(3)mbt[ts1]* and *l(3)mbt[E2]* alleles develop brain tumours that are more invasive and present a much higher rate of developing as immortal neoplasms if they originate in male larvae [14].

In addition to developing malignant brain neoplasms, *l(3)mbt* mutant larvae also exhibit hyperplastic imaginal discs that have not been classified as malignant because in allograft assays they are not lethal to the host and retain the capacity to differentiate [12].

Unlike mbt brain tumours that have been extensively characterised [4,10,12,14,15], little is known about mbt imaginal disc tumours. This is unfortunate because the columnar epithelium of the wing imaginal disc has become a paradigm for the study of epithelia-derived tumours [16,17]. Here we report the results of a study of the hyperplastic *l(3)mbt* mutant wing imaginal discs. We define the mbt wing disc tumour transcriptome and find it to include genes involved in reactive oxygen species (ROS) metabolism. Furthermore, we show the presence of oxidative stress in mbt wing discs, even in apoptosis-blocked conditions. We also find that chemically blocking oxidative stress in mbt wing discs results in partial suppression of the overgrowths. Together, our results show that the *l(3)mbt* is involved in preventing detrimental oxidative stress.

## 2. Materials and Methods

### 2.1. Drosophila Strains and Genotypes

For all the experiments, we used mbt tumours caused by expressing *UAS-l(3)mbt-RNA*i and *UAS-Dcr2* from the ubiquitous *Ubi-Gal4* promoter in individuals that are heterozygous for the *l(3)mbt^ts1^* allele [13], called *mbt-RNA**i*, hereafter. Because of the temperature-sensitive condition of mbt, all crosses, including controls, were maintained at 29 °C.

The following strains were used:

(a) For expression profiling analyses: *w; Ubi-Gal4 UAS-Dcr2/+; UAS-l(3)mbt-RNAi DsRed l(3)mbt^ts1^/+* as mbt-RNAi tumours, and *w; Ubi-Gal4 UAS-Dcr2/+* as control.

(b) To detect ROS in mbt tumours: *w; Ubi-Gal4 UAS-Dcr2/gstD1-GFP; UAS-l(3)mbt-RNAi DsRed l(3)mbt^ts1^/+.* The strain *w; gstD1-GFP/+* and the *w; Ubi-Gal4 UAS-Dcr2/gstD1-GFP* were used as controls.

(c) To inhibit apoptosis in mbt tumours: *w; Ubi-Gal4 UAS-Dcr2/+; UAS-l(3)mbt-RNAi DsRed l(3)mbt^ts1^/UAS-p35*.

(d) To detect ROS after p35 inhibition of apoptosis: *w; Ubi-Gal4 UAS-Dcr2/gstD1-GFP; UAS-l(3)mbt-RNAi DsRed l(3)mbt^ts1^/UAS-p35*.

(e) To genetically decrease ROS levels by overexpression of *Superoxide dismutase 1* (*Sod1*) and *Catalase* (*Cat*): *w; Ubi-Gal4 UAS-Dcr2*/*UAS-Sod1 UAS-Cat; UAS-l(3)mbt-RNAi DsRed l(3)mbt^ts1^*. The strain *w; Ubi-Gal4 UAS-Dcr2*/*UAS-mCD8-GFP; UAS-l(3)mbt-RNAi DsRed l(3)mbt^ts1^* was used as control.

The *gstD1-GFP* strain expressd GFP under control of a 2.7 kb genomic sequence upstream of *gstD1*, an oxidative stress response gene [18]. All other Drosophila strains used in this work were obtained from the Bloomington *Drosophila* Stock Center (BDSC) or the Vienna *Drosophila* Resource Center (VDRC), and are described in [13].

All the experiments were performed using standard food, except in the ROS scavenging experiments. To prevent ROS accumulation, we prepared standard food supplemented with N-acetyl cysteine (Sigma-Aldrich) at a final concentration of 100 μg/mL.

### 2.2. Wing Imaginal Disc Staining and Imaging

Nucleic acid staining was performed by incubating discs for 10 min with 1 μM TO-PRO-3 (Life Technologies, Carlsbad, CA, USA). Phalloidin-Rhodamine Red (Life Technologies) was used at 1:20 dilution for 30 min to label the F-Actin network. For the detection of apoptotic cells, we used the TUNEL assay. We employed the fluorescently labelled Alexa Fluor^®^ 647-aha-dUTP (Thermo Fisher Scientific, Waltham, MA, USA), incorporated using terminal deoxynucleotidyl transferase (Roche, Basel, Switzerland). Immunostainings were performed using standard protocols.

For ROS detection in living tissue, we used CellROX Deep Red Reagent (Life Technologies). Third instar discs were dissected in Schneider’s medium and incubated for 15 min in medium containing 5 μM CellROX Deep Red Reagent, followed by three washes and mounted in culture medium. Samples were protected from light throughout the entire experimental procedure. Images were taken in vivo in a Leica SPE confocal microscope.

Fluorescently labelled secondary antibodies were from Thermo Fisher Scientific. Discs were mounted either in antifade supplemented with TO-PRO3 1:1000 (Thermo Fisher Scientific) to enhance labelled nuclei or in SlowFade mounting media (Thermo Fisher Scientific).

A Zeiss LSM880 and a Leica SPE confocal laser scanning microscope were used for image acquisition. Images were processed and analysed using FIJI software (version 1.53q).

Discs were dissected, labelled, and imaged. The stack images were analysed to search for possible overgrowth and were classified accordingly.

### 2.3. Quantification of Brain Phenotypes

For anatomy analysis, brains from larvae of 7 days after egg laying (dAEL) were dissected, fixed, and labelled with DAPI. The ratios area of neuroepithelium/area of the brain lobe (area NE/BL) and area of central brain/area of the brain lobe (area CB/BL) were calculated by using images acquired with a SP8 Leica confocal image microscope and by measuring the areas corresponding to the NE, the CB, and the brain lobe using FIJI software (version 1.53q). The results were represented in boxplots, and *p*-values were calculated using nonparametric Mann–Whitney *U* tests using GraphPad Prism 9.00 for MacOS X (GraphPad Software, La Jolla, CA, USA) (www.graphpad.com).

### 2.4. Larval Brain Images

Brains were dissected in phosphate-buffered saline (PBS), fixed in 4% formaldehyde, and permeabilised in PBS-0.3% Triton X-100 (PBST). DNA was stained with DAPI. Larval brains were mounted in Vectashield (Vector Laboratories, Newark, NJ, USA). Images were acquired with an SP8 Leica confocal image microscope and processed in Adobe Photoshop CS6 (Adobe, Inc., San José, CA, USA) and ImageJ.

### 2.5. Expression Profiling

For expression analysis, experiments were conducted in triplicates. For each replicate, 10 discs from larvae of 6 dAEL were dissected in Schneider’s Insect medium (Sigma, St. Louis, MO, USA). Discs were homogenized using the pipette in 45 µL of lysis buffer (20 mM DTT, 10 mM Tris-HCl pH 7.4, 0.5% SDS, and 0.5 µg/µL proteinase K). Samples were then incubated at 65 °C for 15 min. RNA was purified using RNA Clean XP bead suspension (Agencourt Bioscience, Beverly, MA, USA), according to manufacturer’s instructions. Microarrays were performed by the IRB Functional Genomics Facility. Briefly, cDNA was generated from 25 ng of RNA using the TransPlex^®^ Complete Whole Transcriptome Amplification Kit (Sigma; reference WTA2) and 17 cycles of amplification. Subsequently, 8 µg of cDNA was fragmented and labelled using GeneChip Human Mapping 250K Nsp Assay Kit (Affymetrix; catalogue # 900766), according to manufacturer’s instructions. cDNA was hybridized to the GeneChip Drosophila Genome 2.0 Array (Affymetrix, (Santa Clara, CA, USA), catalogue # 520087) for 16 h at 45 °C in a GeneChip Hybridization oven 645 (Affymetrix). Washing and staining of microarrays were performed using a GeneChip Fluidics Station 450 (Affymetrix) and arrays were scanned with GeneChip scanner GSC3000 (Affymetrix). Affymetrix GeneChip Command Console software (AGCC) was used to acquire GeneChip images and generate. CEL files for analysis.

### 2.6. Differential Gene Expression Analysis

The microarray analysis protocol from *R* BioConductor [19] workflows was adapted to analyse the samples. CEL files were processed with *oligo* package, which performed background correction using a deconvolution method, quantile normalization, and the Robust Multichip Average (RMA) algorithm for summarization [20]. An extra quality control step was undertaken by applying the Relative Log Expression (RLE) transform [21]. Lowly expressed genes were filtered out from the resulting expression matrix. As microarray data commonly show a large number of probes in the background intensity range, combining a low variance with a low intensity, they could end up being detected as differentially expressed although they are barely above the “detection” limit and are not very informative in general. A “soft” intensity-based filtering was applied as recommended by the *limma* user guide [22]. The row-wise medians from the expression data were computed, as they represent the transcript medians, and a median intensity cut-off was chosen based on the data histogram; this resulted in 6624 transcripts excluded. Gene annotation was retrieved for the corresponding probe identifiers of the GeneChip Drosophila Genome 2.0 Array (Affymetrix, catalogue # 520087), from the *drosophila2.db R* library [23]. Here, only those probes that mapped to an annotation were kept because transcript cluster identifiers can refer to multiple gene symbols and, therefore, they cannot be unambiguously assigned. In summary, 12,043 transcripts were left for the downstream differential gene expression analysis. Different contrasts were considered when computing *limma* linear models, taking into account pair-wise comparisons of male versus female. Gene expression levels were considered as significant when the adjusted *p*-value was below 0.001 and the absolute log fold-change was greater than 1.5. Protein class annotation of those genes was conducted using the Panther web tool [24].

## 3. Results and Discussion

Larvae mutant for the *l(3)mbt* present tumours both in the brain and in imaginal discs [12] and are, therefore, well-suited to investigate the tissue context-dependent effect of tumour suppressor loss. The tumours that result from the loss of *l(3)mbt* function in the brain have been studied in detail [4,10,13,14]. In this study, we have sought to investigate *l(3)mbt* loss-of-function tumorous imaginal discs.

To this aim, we analysed discs that drive the *UAS-l(3)mbt-RNAi* under the *Ubi-Gal4* in flies that are heterozygous for the *l(3)mbt[ts1]* allele. We have found that wing discs from late *mbt-RNAi* third instar larvae present different levels of overgrowth that can be assigned to one of three distinct phenotypic classes (Figure 1A): (1) discs without any morphological anomaly; (2) discs with mild effects consisting of small overgrowth, primarily in the hinge zone; (3) discs with strong defects consisting of large or multiple overgrowths, not only in the hinge but also in other regions of the wing disc, such as the pouch or the notum. Our results showed that 52% of the female larvae and 63% of the male larvae showed overgrown discs (Classes 2+3) (Figure 1B). The incidence of the strong overgrowth phenotype (Class 3) is also slightly higher in male (32%) than in female (18%) larvae (Figure 1B). Previous analyses have suggested that the hinge is an “oncogenic hotspot” in the wing imaginal disc [25,26,27,28]. Our observation that Class 2 and 3 phenotypes have a propensity to develop overgrowths in the hinge substantiates this hypothesis. These results confirm the presence of tumorous discs in *l(3)mbt* mutant larvae [12]. They also reveal that in stark contrast with the strong sex-dimorphic traits presented by mbt brain tumours in *l(3)mbt[ts1]* and *l(3)mbt[E2]* homozygous and transheterozygous larvae [14], sex dimorphism is only marginal in the case of *mbt-RNAi* disc tumours. To determine if such differences could be due to the mutant condition (i.e., *mbt-RNAi* versus classical mutant alleles), we studied *mbt-RNAi* brain tumours and found that sex-dependent differences are also marginal, if any exist, with most male and female mbt-RNAi brain tumours closely resembling those found in male *l(3)mbt[ts1]* and *l(3)mbt[E2]* homozygous and transheterozygous larvae (Figure 1C,D).

To further characterise mbt tumorous wing discs we carried out a genome-wide transcriptome analysis of *mbt-RNAi* and control (*w; Ubi-Gal4 UAS-Dcr2/+*) wing discs. We identified a total of 241 and 283 genes differentially expressed (DE) in female and male *mbt-RNAi* samples, respectively, compared to control samples (Appendix A). Consistent with the very low extent of sex dimorphism observed in disc tumour anatomy, we found that most dysregulated genes were so in both sexes (*n* = 214). The major effect observed on the mbt wing disc transcriptome is the up-regulation of gene expression (Figure 2A,B).

Our data show that among the 100 genes that belong to the mbt tumour signature (MBTS) identified in mbt brain tumours or allograft transplantations [10], 50 and 54 genes were also misregulated in female and male *mbt-RNAi* discs, respectively (Appendix A). A quarter of the MBTS genes are known to normally function in the germline [10]. Notably, we also found 17 and 18 germline genes up-regulated in female and male mbt wing discs, respectively (Appendix A).

Functional annotation of the dysregulated genes in our transcriptome profiling study shows an enrichment in proteins belonging to the oxidoreductase (10 DE genes) and metabolite interconversion enzyme (19 DE genes) classes (Figure 2C; Appendix A) with some genes belonging to both categories. Included among the up-regulated genes belonging to those categories are *Peroxinectin-like* (*Pxt*), the thioredoxin testis-specific *Thioredoxin T* (*TrxT*), and the ovary-specific thioredoxin *deadhead* (*dhd*). All three genes belong to the mbt brain tumour signature MBTS [10]. These results reveal that the loss of *l(3)mbt* in wing discs brings about the dysregulation of the transcription of genes with functions that include oxidation-reduction (redox) processes. Indeed, ROS are involved in the development and progression of cancer types and an increased ROS level is considered a hallmark of many tumours.

To investigate whether ROS is produced in mbt tumours, we monitored the expression of *gstD1-GFP*. The *gstD1-GFP* reporter expresses GFP under the control of a 2.7 kb genomic sequence upstream of the oxidative stress response gene *gstD1*, which is an efficient tool to track ROS in tissues [18]. In non-stressed wild-type discs, we found *gstD1-GFP* to be expressed in the peripodial membrane, a squamous epithelial layer which covers the apical side of the columnar epithelium of the disc and the laterals that link the columnar epithelium with the peripodial membrane [29] (Figure 3A). At the proper columnar epithelium, only the dorsoventral boundary of the wing pouch showed a GFP signal (Figure 3B). Notably, *gstD1-GFP* is strongly up-regulated in *mbt-RNAi* wing discs, with the highest GFP signal located in the hinge region (Figure 3C). In mbt larval brains, however, we found no evidence of *gstD1-GFP* expression in the tumour tissue that spreads over the lamina, neuroepithelium, medulla, and central brain (Figure 3E). In fact, *gstD1-GFP* expression levels are indistinguishable between *mbt-RNAi* and wild-type brain lobes, both of which present GFP fluorescence levels that are low and restricted to glial cells enveloping the lobes. We also detected ROS production in *mbt-RNAi* wing imaginal discs using the ROS-sensitive dye CellROX Deep Red. Wild-type discs show only low levels of CellROX in the margins. In contrast, *mbt-RNAi* showed extensive staining in many cells of the disc (Appendix A).

These results reveal a significant build-up of ROS in mbt imaginal disc tumours, but not in mbt brain tumours. ROS induction, as well as a dependence upon ROS, have been extensively documented in a wide range of tumour models in larval *Drosophila* imaginal discs and have been considered to be common traits of epithelial tumours, independently of their origin [30,31]. The case of mbt tumours reported here is fully consistent with ROS build-up being a common trait of imaginal disc tumours, but not of epithelial tumours in general because mbt brain tumours derive from the neural epithelium [4].

Our transcriptomic profiling analysis together with the expression of *gstD1-GFP* observed in mbt discs suggest a role of L(3)mbt in ROS metabolism. However, it is known that cells entering apoptosis produce ROS, likely through mitochondrial disruption [32]. For this reason, we study the contribution of apoptosis to the *mbt-RNAi* wing phenotype. As a first step, we carried out immunofluorescence for the detection of apoptotic cells with TUNEL assay. In mbt wing discs, we detected a high concentration of apoptotic cells in the hinge, which coincides with the *gstD1-GFP* accumulation (Figure 4A). In very late third instar mbt discs, high levels of *gstD1-GFP* are present, and apoptotic cells spread through the disc, together with a general disorganisation of the epithelium (Figure 4B). Together, these results indicate that ROS is accumulated progressively in *mbt-RNAi* discs concomitantly with an increase in apoptosis.

We then analysed the phenotype of *mbt-RNAi* discs ectopically expressing the baculovirus protein p35 which blocks the function of the effector caspases [33]. Upon p35 expression, apoptotic cells that would normally be removed from the tissue remain alive in the epithelium. This assay facilitates the evaluation of the overgrowth potential. We found more extensive hyperplastic tissue overgrowth in the *p35 mbt-RNAi* than in the *mbt-RNAi* discs, with overgrowths distributed all over the wing disc epithelium (Figure 4C–E; *n* = 40). These findings suggest that overgrowth in *mbt-RNAi* discs is limited due to the loss of cells by apoptosis.

Remarkably, the analysis of *gstD1-GFP* accumulation in *p35 mbt-RNAi* discs reveals the existence of overgrowths associated with ROS production (n = 92) (Figure 4F–H). We found the *gstD1-GFP* expression to be highly accumulated in the entire disc, coinciding with the epithelial folding resulting from the overgrowth (Figure 4H). It has been reported that the production of ROS could be due to mitochondrial dysfunction induced by damaged cells [34]. However, our results suggest that mbt mutant cells are able to produce ROS in the absence of apoptosis. This indicates that ROS production is not only a consequence of apoptosis, but also a key feature of *mbt-RNAi* mutant discs. Additionally, we observed a more extensive expression of *gstD1-GFP* in *p35 mbt-RNAi* compared to *mbt-RNAi*, which suggests that the prevention of cell elimination by apoptosis generates a more extensive ROS-producing mutant tissue.

Having found that mbt mutant epithelial cells exhibit high ROS levels, we asked whether *l(3)mbt* protects from detrimental oxidative stress. To answer this question, we investigated the effect of the exposure of mbt mutant larvae to antioxidants, which should mitigate the mbt wing disc phenotype. We fed *mbt-RNAi* larvae with food containing the ROS scavenger N-acetyl cysteine (NAC) and checked for incidences of overgrowth in comparison to standard food. Larvae raised in NAC-supplemented food resulted in a remarkable drop in the incidences of overgrowths in the imaginal discs in comparison to *mbt-RNAi* discs from larvae grown in standard food (Figure 5A). Low levels of GFP were still detected in morphologically normal *mbt-RNAi* discs, which suggests that the blockade of oxidative stress by NAC is partial (Figure 5B,C). In those NAC-reared animals in which overgrowths are still present, the high *gstD1-GFP* was associated with the overgrowth (Figure 5D,E). In addition, the expression of the ROS scavengers *Superoxide dismutase 1* (*Sod1*) and *Catalase* (*Cat*) genes in a *mbt-RNAi* background resulted in a partial recovery of the wing disc phenotype in female larvae (Appendix A). Together, these observations showed that ROS scavenging is able, albeit partially, to recover the normal morphology of mbt discs.

In this work, we have re-evaluated the potential of *l(3)mbt* mutants to generate overgrowth in wing imaginal discs as firstly observed by Gateff and colleagues [12]. We also have shown that *l(3)mbt* mutant wing discs have an altered expression profile, which included a set of MBTS genes. Among the dysregulated genes in mbt mutant discs, we found genes involved in the redox balance of the cell which suggests that *l(3)mbt* has a role in preventing damage by oxidative stress. Indeed, *l(3)mbt* mutant epithelia show the ROS accumulation detected by the *gstD1-GFP* reporter associated with overgrowth. Moreover, the reduction in the oxidative stress by feeding the animals with the NAC antioxidant results in the amelioration of the tumour progression which demonstrates the involvement of oxidative stress in *l(3)mbt* mutants. In addition, the different response in terms of ROS accumulation observed in mbt mutant brains and wing discs documents another instance in which a given tumorigenic event (i.e., loss of *l(3)mbt* function) triggers different tumour growth pathways depending on the tissular context. We do not know what the basis for the different ROS responses in mbt wing discs and brain tumours may be. The set of dysregulated redox-related genes is rather similar in both tumours, hence suggesting that the microenvironment of each tissue may be crucial to determine the stress response in each organ. Indeed, differences in ROS production have also been observed upon exposure to ionizing irradiation that induces high levels of apoptosis in imaginal discs [35,36] but not in larval brains [37]. Interestingly, our transcriptome profiling study showed two upregulated genes that could have a role in the apoptotic phenotype of mbt-stressed wing discs. The first is the Cdk5alpha subunit of Cdk5, which phosphorylates Mekk1 that activates JNK, which in turn triggers apoptosis [38]. The second is p53, the sole *Drosophila* member of the p53 family, that in response to stress initiates apoptosis by activating the transcription of the pro-apoptotic gene, *reaper* [39,40,41,42].

## Figures and Tables

**Figure 1 cells-11-02542-f001:**
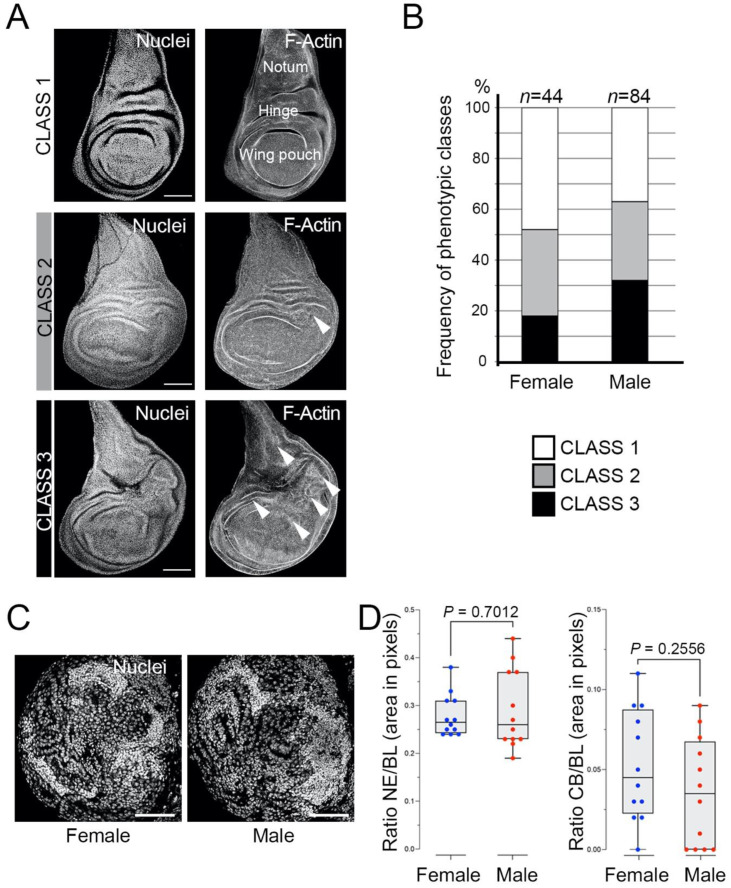
*mbt-RNAi* mutants produce overgrowth in wing imaginal discs and brains. (**A**) Wing discs from *mbt-RNAi* mutant larvae stained with TO-PRO3 (Nuclei) and Phalloidin (F-actin), showing the three phenotypic classes: (1) absence of overgrowth; (2) mild and few overgrowths; (3) strong and multiple overgrowths. Arrowheads point to anomalous folding resulting from multiple overgrowths. The three main regions of the disc (wing pouch, hinge, and notum) are indicated. Scale bars: 50 µm. (**B**) Incidence of the overgrowth phenotypes in female and male *mbt-RNAi* wing discs. (**C**) Larval brain lobes from female and male *mbt-RNAi* larvae stained with DAPI (Nuclei). Female and male brain lobes present reduced central brains (CBs) and overgrown neuroepithelia (NE). Scale bars: 50 µm. (**D**) Relative sizes of NE and CB (as a fraction of brain lobe (BL) area) in female (blue) and male (red) *mbt-RNAi* mutant larvae. No significant differences are observed in NE and CB sizes between mbt female and male brain lobes.

**Figure 2 cells-11-02542-f002:**
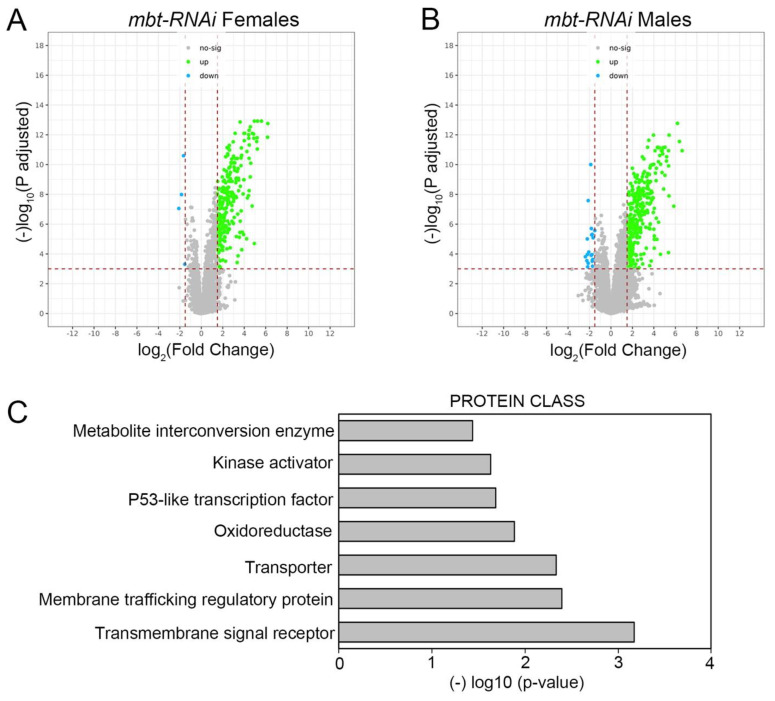
Volcano plots showing the change in expression of each individual gene in females (**A**) and males (**B**) in *mbt-RNAi* tumours compared to control. Differentially down-regulated and up-regulated genes (|FC| > 1.5; adjusted *p*-value < 0.001) are indicated in cyan and green, respectively. (**C**) Protein class enrichment for the set of DE genes in *mbt-RNAi* tumours.

**Figure 3 cells-11-02542-f003:**
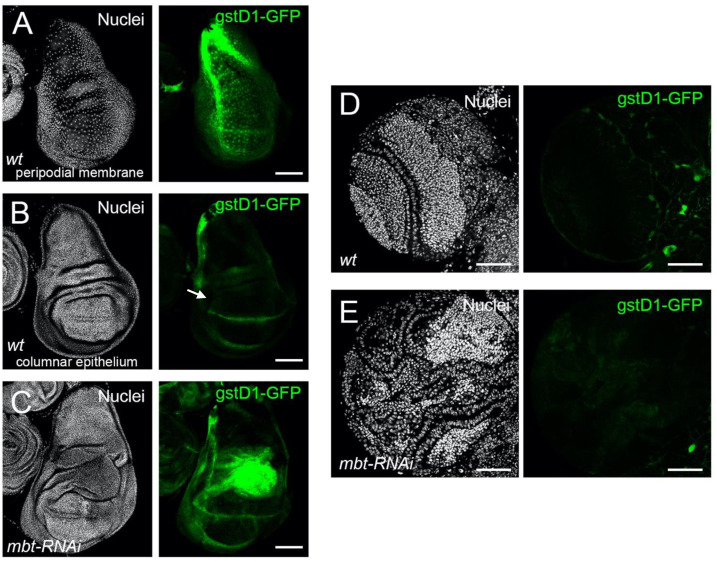
Oxidative stress detected with *gstD1-GFP* reporter in *mbt-RNAi* mutant wing imaginal discs and brain lobes. (**A**–**C**) Wing imaginal discs and (**D**,**E**) brain lobes stained with DAPI (nuclei in gray) and expressing the ROS reporter *gstD1-GFP* (green). (**A**) Wing discs from wild-type larvae focusing on the peripodial membrane. (**B**) Same disc showing the columnar epithelium; the arrow points to the DV boundary. (**C**) Wing disc from *mbt-RNAi* mutant larvae, with strong *gstD1-GFP* signal in the hinge region. Brain lobes from wild-type (**D**) and *mbt-RNAi* larvae (**E**) showing similar background levels of *gstD1-GFP* signal (green). Scale bars: 50 µm.

**Figure 4 cells-11-02542-f004:**
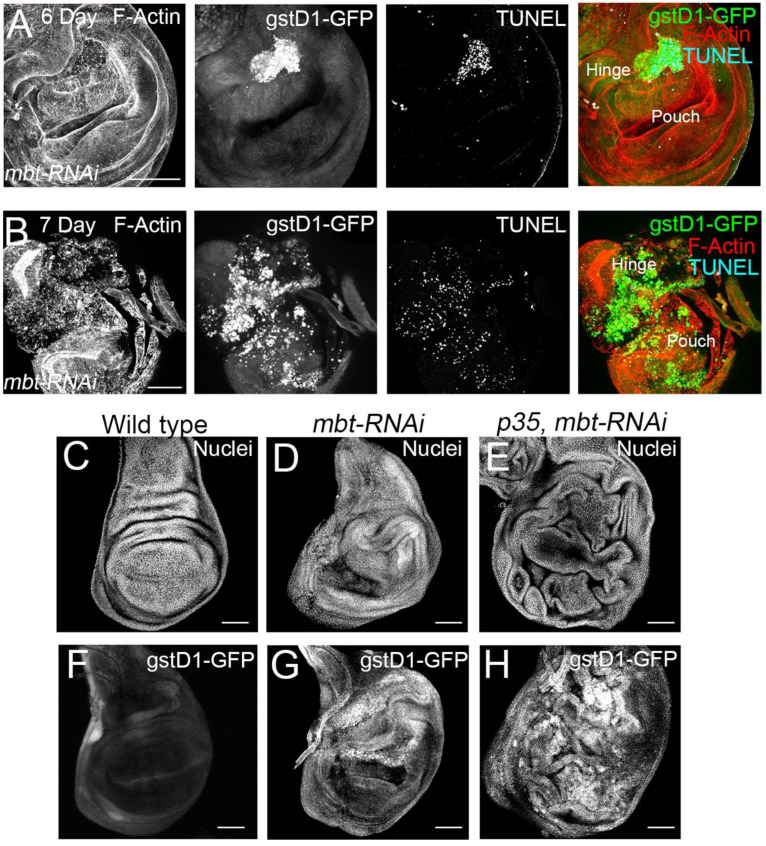
Apoptosis and ROS production in *mbt-RNAi* wing discs. (**A**,**B**) Wing disc from *mbt-RNAi* mutant larvae of 6 dAEL (**A**) and 7 dAEL (**B**). Expression of *gstD1-GFP* is shown in green, apoptosis is revealed by TUNEL (blue), and F-actin is stained with Phalloidin (red). Note the increase in severity of the overgrowths and extension and intensity of *gstD1-GFP* signal. (**C**–**H**) Wing discs stained with TOPRO (Nuclei; **C**–**E**) and showing *gstD1-GFP* expression (**F**–**H**) from wild-type (**C**,**F**), *mbt-RNAi* (**D**,**G**), and *mbt-RNAi* co-expressing p35 larvae (**E**,**H**). Co-expression of p35 enhances the overgrowths and *gstD1-GFP* is still present. Scale bars: 50 µm.

**Figure 5 cells-11-02542-f005:**
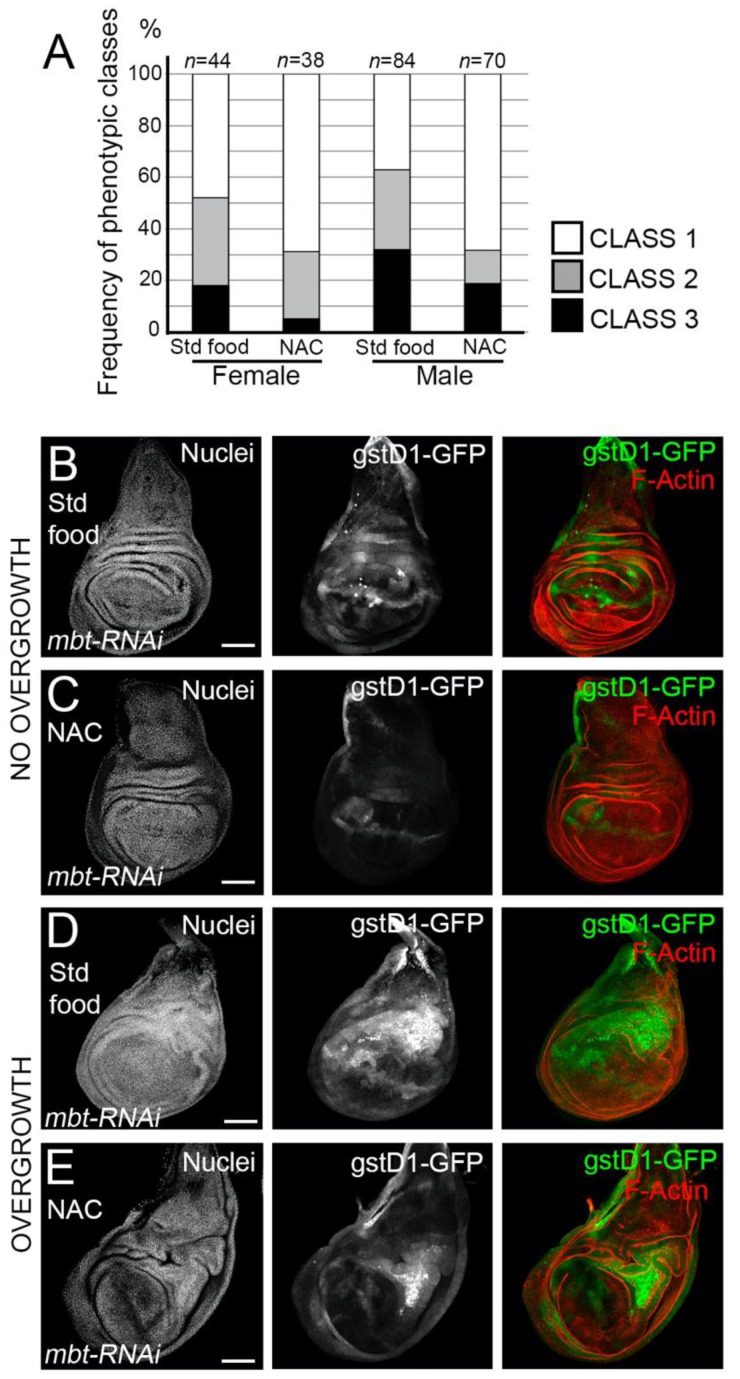
Effects of the ROS scavenger NAC on the *mbt-RNAi* phenotype. (**A**) Incidence of discs with overgrowth in *mbt-RNAi* fed with standard food or with food supplemented with NAC in females and males. (**B**,**C**) *mbt-RNAi* wing discs with no folding or overgrowths, but weak *gstD1-GFP* activation grown in standard food (**B**) and grown in food supplemented with NAC (**C**). (**D**,**E**) *mbt-RNAi* wing discs with severe folding and overgrowths with strong *gstD1-GFP* expression grown in standard food (**D**) and grown in food supplemented with NAC €. Discs were stained with TO-PRO3 (Nuclei; gray) and Phalloidin (F-actin; red). Scale bars: 50 µm.

## Data Availability

All data needed to evaluate the conclusion in the paper are present in the manuscript and/or the Appendix A. Additional data related to this paper may be requested from the authors.

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
