# Peer review of "Oxidative Stress Is Associated with Overgrowth in Drosophila l(3)mbt Mutant Imaginal Discs"

_cells, 2022, doi:10.3390/cells11162542_

Round 1

Reviewer 1 Report

In this MS, Climent-Cantø and cols. analyse mbt tumors formed in the imaginal discs. Mbt is a classic tumor model that has been extensively analysed by the González group. Previous analyses by this group were aimed at understanding how those mutants drive brain tumor formation. In this analysis, the authors evaluate the molecular basis driving tumor formation in mbt wing imaginal discs. They analyse the expression profile of those tumors, which reveal the deregulation of genes in redox processes. Consistently, those tumors show an increase in ROS, which is a common trait in fly and human tumors. The authors also performed chemically based functional analyses that show that ROS production is a contributing factor in mbt tumor formation in the imaginal discs. The topic analysed here is very interesting. He results are presented in a clear manner. The conclusions are well supported by the results. Finally, I would like to stress that the paper is exceptionally well structured and written. This facilitates greatly the understanding of this study even to readers that are not experts in this field.

Although the paper could be published in its current version, I have some suggestions that the authors might consider and that, in my opinion, might increase the interest and impact of this study. See these specific comments below:

Line 159: ”We have found that wing discs from late mbt-RNAi third instar larvae present different levels of overgrowth that can be assigned to one of 3 distinct phenotypic classes (Fig 1A)”  I would suggest indicating in the text which is the Gal4 driver used for that analysis. This will facilitate the understanding of the experimental approach to the reader.

Fig 1A. The class 3 phenotype is referred to in the text as: “discs with strong defects consisting in large or multiple overgrowths not only in the hinge but also in other regions of the wing disc such as the pouch or the notum”.  When looking at the image provided in that Fig, I can observe that the phenotype is more severe than the one presented as class 2, but it seems to me that the overgrowth is still restricted to or originating from the hinge area. The folds observed in the notum could be the consequence of the overgrowth in the hinge. I personally find this observation very interesting because previous analyses have suggested that the hinge is an “oncogenic hotspot” in this organ. Some of those studies are, Sander et al., 2018; Khan et al., 2013; Tamori and Deng, 2017; Tamori et al., 2016. The results presented in this MS could provide additional evidence supporting that idea. I would suggest the authors to consider this and maybe mentioning and briefly discussing it.

I found very interesting the observation that ROS was strongly induced in the mbt wing disc tumors but not in brain tumors. Are the genes involved in redox processes upregulated in both mbt wing disc and mbt brain tumors? This could help to explain the difference in this trait between both models. The authors might consider mentioning and/or discussing this aspect during this revision.

Author Response

Reviewer #1

Line 159: ”We have found that wing discs from late mbt-RNAi third instar larvae present different levels of overgrowth that can be assigned to one of 3 distinct phenotypic classes (Fig 1A)”  I would suggest indicating in the text which is the Gal4 driver used for that analysis. This will facilitate the understanding of the experimental approach to the reader.

The requested information was shown in the Materials and Methods, but the reviewer is right in that it may be helpful to indicate it also in the corresponding paragraph of the Results section. To this end we have introduced the following sentence:

"In this study we have sought to investigate l(3)mbt loss-of-function tumorous imaginal discs. To this aim, we analysed discs that drive the UAS-l(3)mbt-RNAi under the Ubi-Gal4 in flies that are heterozygous for the l(3)mbt[ts1] allele.”

Fig 1A. The class 3 phenotype is referred to in the text as: discs with strong defects consisting in large or multiple overgrowths not only in the hinge but also in other regions of the wing disc such as the pouch or the notum”.  When looking at the image provided in that Fig, I can observe that the phenotype is more severe than the one presented as class 2, but it seems to me that the overgrowth is still restricted to or originating from the hinge area. The folds observed in the notum could be the consequence of the overgrowth in the hinge. I personally find this observation very interesting because previous analyses have suggested that the hinge is an oncogenic hotspot” in this organ. Some of those studies are, Sander et al., 2018; Khan et al., 2013; Tamori and Deng, 2017; Tamori et al., 2016. The results presented in this MS could provide additional evidence supporting that idea. I would suggest the authors to consider this and maybe mentioning and briefly discussing it.

We thank the reviewer for raising this issue. Indeed, there is a propensity to develop a phenotype in the proximal regions of the wing disc. As suggested by the reviewer we have rephrased the text to refer to this important issue. The revised version is as follows:

…..also slightly higher in male (32%) than in female (18%) larvae (Figure 1B). “Previous analyses have suggested that the hinge is an “oncogenic hotspot” in the hinge of the wing imaginal disc (disc (Khan et al., 2013; Sander et al., 2018; Tamori et al., 2016; Tamori and Deng 2017). Our observation that Class 2 and 3 phenotypes have a propensity to develop overgrowths in the hinge substantiates this hypothesis.”

I found very interesting the observation that ROS was strongly induced in the mbt wing disc tumors but not in brain tumors. Are the genes involved in redox processes upregulated in both mbt wing disc and mbt brain tumors? This could help to explain the difference in this trait between both models. The authors might consider mentioning and/or discussing this aspect during this revision.

As requested by the reviewer, we have added a short comment discussing this point:

“We do not know what the basis for the different ROS response in mbt wing disc and brain tumors may be. The set of disregulated redox-related genes is rather similar in both tumours, hence suggesting that the microenvironment of each tissue may be crucial to determine the stress response in each organ. Indeed differences in ROS production have also been observed upon exposure to ionizing irradiation that induces high levels of apoptosis in imaginal discs (Pérez_Garijo et al 2004; Milán et al., 1997) but not in larval brains (Wagle and Song 2020).”

Reviewer 2 Report

In this work, the authors study a tumour model caused by the loss of the tumour suppressor l(3)mbt. This model has previously been extensively characterised in Drosophila brains, where it causes malignant tumours. Here, they investigate the not so thoroughly characterised mbt wing imaginal discs phenotype, which also cause hyperplastic growth.

The authors show that, upon knockdown of mbt, around 50% of the wing discs overgrow. They then carry out a transcriptome analysis and observe a general up-regulation of genes in mbt-RNAi discs. Among the upregulated genes, there are genes with redox functions that belong to the mbt tumour signature. Indeed, they use an oxidative stress reporter line and observe high ROS in the tumour wing discs. However, ROS is not upregulated in the tumour brains. Those tumour wing discs with high ROS are also positive for apoptotic markers. Interestingly, upon inhibition of apoptosis, they observe more hyperplastic overgrowth and higher ROS accumulation in the tumours, suggesting apoptosis limits the tumour from becoming more malignant. Finally, they fed the flies with the ROS scavenger N-acetyl cysteine (NAC) and observe a reduction in tumour growth upon treatment, proving a role of oxidative stress in mbt tumour wing discs.

This study demonstrate that tissue context is critical for malignant growth and performs a nice characterisation of mbt tumour signature for the wing discs.

Major comments:

1.     Did the mbt-RNAi discs used for RNAseq belong to Class I, II or III, or were they randomly selected?

2.     There seems to be more gstd-GFP upregulated upon p35 overexpression. The gstd-gfp upregulation should be quantified. Is there an interaction between these two pathways?

3.     Are any apoptotic genes up-regulated in mbt-RNAi transcriptome analysis?

4.     From this work, it is evident that apoptosis is upregulated in the tumour and that it limits the tumour from growing even more. However, this was not investigated in the brain. Is there more apoptosis in the tumour brains? If not, is that the reason why those brain tumours are more malignant? If you increase oxidative stress in the tumour brain, would you expect to see higher apoptosis and therefore a reduction in tumour size?

5.     Reduction of oxidative stress ameliorates tumour growth. What is the mechanism behind it? Is knockdown of other up-regulated genes such as Pxt, TrxT or dhd sufficient to reduce hyperplasia?

6.     I would like to see a more extensive discussion on the differences between the brain tumours and the wing discs tumours.

Minor comments

1.     Line 188: specify the upregulated genes are germline-related

2.     Line 194: …multiple overgrowths.

3.     Line 201: “Functional annotation of the dysregulated genes in our transcriptome …”

4.     Line 217: missing references

5.     Line 242: “ROS induction, as well as dependence upon ROS, has been extensively…” adding commas helps with reading flow

6.     Line 243: Drosophila in italics

7.     Line 260. Revise “Under expression of p35, apoptotic cells that normally should be removed from the tissue will remain alive in the epithelium and is used as an assay to evaluate the potential of overgrowth.”

8.     Line 313: missing comma “In addition, the different response…”

Author Response

Reviewer #2

Major comments:

  1. Did the mbt-RNAi discs used for RNAseq belong to Class I, II or III, or were they randomly selected?

The discs used for the expression profiling study were randomly selected because it is technically impossible to classify them under the stereomicroscope.

  1. There seems to be more gstd-GFP upregulated upon p35 overexpression. The gstd-gfp upregulation should be quantified. Is there an interaction between these two pathways?

We thank the reviewer for raising this issue. We agree on the importance of quantification and tried to do so by measuring pixel intensity. However, we found that the excess of p35-induced folding makes it impossible to objectively quantify the extent of gstd-GFP upregulation which is nonetheless conspicuous. Consequently, we refer to this observation in qualitative, rather than quantitative terms as follows:

Line 299: ”Additionally, we observed more extensive expression of gstD1-GFP in p35 mbt-RNAi compared to mbt-RNAi, which suggests that preventing of cell elimination by apoptosis generates a more extensive ROS-producing mutant tissue.”

  1. Are any apoptotic genes up-regulated in mbt-RNAi transcriptome analysis?

This is a very interesting question. Two of the genes upregulated in our mbt-RNAi transcriptome analysis are the Cdk5alpha subunit of Cdk5 and p53. Cdk5 phosphorylates Mekk1 that activates JNK that in turn triggers apoptosis (Kang et al. 2012). p53 initiates apoptosis in response to stress by activating transcription of the pro-apoptotic gene reaper (Brodsky et al., 2000; Fan et al., 2010; Wells et al., 2006; Mollerau and Ma 2014). It is likely that upregulation of these two genes may have a role in the apoptotic phenotype of mbt stressed discs.

The revised version includes the following comment on this issue:

“Interestingly, our transcriptome profiling study showed two upregulated genes that could have a role in the apoptotic phenotype of mbt stressed discs. The first is the Cdk5alpha subunit of Cdk5, which phosphorylates Mekk1 that activates JNK, which in turn triggers apoptosis (Kang et al. 2012 ). The second is p53, the sole Drosophila member of the p53 family, that in response to stress initiates apoptosis by activating transcription of the pro-apoptotic gene reaper (Brodsky et al., 2000; Fan et al., 2010; Wells et al., 2006; Mollerau and Ma 2014). “

We thank the reviewer for raising this issue and giving us the opportunity to add this point to the discussion.

  1. From this work, it is evident that apoptosis is upregulated in the tumour and that it limits the tumour from growing even more. However, this was not investigated in the brain. Is there more apoptosis in the tumour brains? If not, is that the reason why those brain tumours are more malignant? If you increase oxidative stress in the tumour brain, would you expect to see higher apoptosis and therefore a reduction in tumour size?

This is an interesting point as differences in brains and discs are evident. We hardly see any evidence of apoptosis and ROS in brains. We currently do not have evidence that increasing stress in would ameliorate tumorigenesis in brains.

  1. Reduction of oxidative stress ameliorates tumour growth. What is the mechanism behind it? Is knockdown of other up-regulated genes such as Pxt, TrxT or dhd sufficient to reduce hyperplasia?

We thank this reviewer for the suggestion. We do not have those experiments, but rescue experiments with a particular set of candidates will be done in the future.

  1. I would like to see a more extensive discussion on the differences between the brain tumours and the wing discs tumours.

As requested (also by reviewer 1) we have expanded the discussion on this issue

Minor comments

  1. Line 188: specify the upregulated genes are germline-related

Done (now line 208).

  1. Line 194: …multiple overgrowths.

Done (now line 214).

  1. Line 201: “Functional annotation of the dysregulated genes in our transcriptome …”

Done (now line 228).

  1. Line 217: missing references

We have added a recent, very comprehensive review on the subject (Morata and Calleja 2020) (now Line 243).

  1. Line 242: “ROS induction, as well as dependence upon ROS, has been extensively…” adding commas helps with reading flow

Done (now line 267).

  1. Line 243: Drosophila in italics

Done (now line 268).

  1. Line 260. Revise “Under expression of p35, apoptotic cells that normally should be removed from the tissue will remain alive in the epithelium and is used as an assay to evaluate the potential of overgrowth.”

The sentence has been revised: (now line 285)

“Upon p35 expression, apoptotic cells that would normally be removed from the tissue remain alive in the epithelium. This assay facilitates the evaluation of overgrowth potential".

  1. Line 313: missing comma “In addition, the different response…”

Done (now line 349).

Reviewer 3 Report

The Drosophila l(3)mbt acts as a tumor suppressor and loss of l(3)mbt has been associated with malignant transformation of the neuroblasts in larval brain. In this manuscript, the authors investigate the role of l(3)mbt in wing development. They found that l(3)mbt mutants exhibit elevated ROS levels and wing disc overgrowth, indicating the involvement of oxidative stress in l(3)mbt wing discs hyperplastic growth. While their findings are interesting, the data presented are still preliminary to support their conclusion.

Major:

1.      The authors found that gstD1-GFP is strongly up-regulated in mbt-RNAi wing discs, thus suggesting an increased ROS production. However, additional ROS indicators such as DHE or H2DCF-DA should be used to confirm their findings.

2.      The high ROS levels in mbt mutants could be derived from mitochondria or ROS producing enzymes such as Duox. Does knockdown of Duox or overexpressing ROS-removing enzymes such as catalases and superoxide dismutases (SOD) rescue mbt mutant phenotypes?

3.      It has been reported that caspases promote overgrowth by inducing the generation of ROS. Does caspase also play a role in mbt mutants?

4.      The JNK signaling has been associated with ROS and cell death. Is JNK pathway also involved in mbt-mediated ROS metabolism?

Minor

1.      The authors should provide the source of the flies (including Gal4 lines and reporters) they used in this study.

Author Response

Reviewer # 3

Major:

  1. The authors found that gstD1-GFP is strongly up-regulated in mbt-RNAiwing discs, thus suggesting an increased ROS production. However, additional ROS indicators such as DHE or H2DCF-DA should be used to confirm their findings.

As requested, we have illustrated ROS with an additional reporter, CellROX Deep Red, that efficiently emits fluorescence in the presence of general ROS in living tissues.

We refer to this result in the revised version as follows:

“We also detected ROS production in mbt-RNAi wing imaginal discs using the ROS-sensitive dye CellROX Deep Red. Wild type discs show only low levels of CellROX in the margins. In contrast, mbt-RNAi showed extensive staining in many cells of the disc (Supplementary Figure S1).”

  1. The high ROS levels in mbt mutants could be derived from mitochondria or ROS producing enzymes such as Duox. Does knockdown of Duox or overexpressing ROS-removing enzymes such as catalases and superoxide dismutases (SOD) rescue mbt mutant phenotypes?

This is an important point that we have tried to address. We have found that lowering ROS by driving Sod1 and Catalase from Ubi-Gal4 appears to partially ameliorate the mutant mbt phenotype (new Supplementary Figure S1).

We refer to this result in the revised version as follows:

"In addition, expression of the ROS scavengers Superoxide dismutase 1 (Sod1) and Catalase (Cat) genes in a mbt-RNAi background resulted in a partial recovery of the wing disc phenotype in female larvae (Supplementary Figure S1).”

Material and Methods has also been modified accordingly.

  1. It has been reported that caspases promote overgrowth by inducing the generation of ROS. Does caspase also play a role in mbt mutants?

Apoptotic caspases contribute to tumor formation in a loop that includes ROS, JNK and macrophages (Pérez et al 2017 doi: 10.7554/eLife.26747). We know that caspases are activated, but we do not have any evidence that supports this role in mbt discs.

  1. The JNK signaling has been associated with ROS and cell death. Is JNK pathway also involved in mbt-mediated ROS metabolism?

Unfortunately, we do not have data on this issue other than the fact that Cdk5 subunit alpha is upregulated in mbt discs. This subunit is responsible for the activation of JNK-dependent apoptosis. Therefore, it is likely that JNK may be involved in mbt-mediated ROS metabolism.

Minor

  1. The authors should provide the source of the flies (including Gal4 lines and reporters) they used in this study.

The requested information has been provided in Material and Methods:

“The gstD1-GFP strain expresses GFP under the control of a 2.7 kb genomic sequence upstream of gstD1, an oxidative stress response gene (Sykiotis and Bohman 2008). All other Drosophila strains used in this work were obtained from Bloomington Drosophila Stock Center (BDSC) or Vienna Drosophila Resource Center (VDRC) and are described in Rossi et al., 2017.”

Round 2

Reviewer 3 Report

The authors have addressed my comments.